# A Novel Fabrication of Hematite Nanoparticles via Recycling of Titanium Slag by Pyrite Reduction Technology

**DOI:** 10.3390/nano14161330

**Published:** 2024-08-08

**Authors:** Genkuan Ren, Yinwen Deng, Xiushan Yang

**Affiliations:** 1Department of Materials and Chemical Engineering, Yibin University, Yibin 644000, China; rgk2000@163.com; 2Chemical Science and Engineering College, Sichuan University, Chengdu 610065, China; 2018413002@yibinu.edu.cn

**Keywords:** reduction method, hematite nanoparticles, titanium slag, pyrite

## Abstract

An enormous quantity of titanium slag has caused not merely serious environment pollution, but also a huge waste of iron and sulfur resources. Hence, recycling iron and sulfur resources from titanium slag has recently been an urgent problem. Herein, hematite nanoparticles were fabricated through a pyrite reduction approach using as-received titanium slag as the iron source and pyrite as the reducing agent in an nitrogen atmosphere. The physicochemical properties of the hematite nanoparticles were analyzed using multiple techniques such as X-ray diffraction pattern, ultraviolet–visible spectrophotometry, and scanning electron microscopy. The best synthesis conditions for hematite nanoparticles were found at 550 °C for 30 min with the mass ratio of 14:1 for titanium slag and pyrite. The results demonstrated that hematite nanoparticles with an average particle diameter of 45 nm were nearly spherical in shape. The specific surface area, pore volume, and pore size estimated according to the BET method were 19.6 m^2^/g, 0.117 cm^3^/g, and 0.89 nm, respectively. Meanwhile, the fabricated hematite nanoparticles possessed weak ferromagnetic behavior and good absorbance in the wavelength range of 200 nm-600 nm, applied as a visible light responsive catalyst. Consequently, these results show that hematite nanoparticles formed by the pyrite reduction technique have a promising application prospect for magnetic material and photocatalysis.

## 1. Introduction

Titanium dioxide (TiO_2_), as an important inorganic chemical material, is widely applied in chemical fibers, plastics, cosmetics, dyestuff, photocatalysis, and other related fields because of its excellent chemical stability, high achromic ability, nontoxicity, and outstanding heat resistance [1,2,3,4]. With the rapid development of the aforementioned fields in recent years, the output and consumption of titanium dioxide have escalated dramatically, with its total production in China exceeding 3.89 million tons in 2021 and 4.16 million tons in 2023 [5,6]. Currently, the primary production technologies of titanium dioxide include the sulfuric acid process as well as the chlorination process [7]. Compared with the chlorination process, the sulfuric acid process is the more attractive technology with a simple and mature process, stable product quality, and low requirement on the grade of the titanium source. Therefore, more than 90% of titanium dioxide is produced through sulfate technology in China. For the sulfuric acid process, the limitation in its sustainable development is that a tremendous quantity of waste such as spent sulfuric acid and titanium slag are generated. As reported, producing a ton of titanium dioxide discharges roughly 3–4 tons of titanium slag with FeSO_4_·7H_2_O being the major component [8], and the annual emission of titanium slag was roughly 7.0 million tons in 2022 and 7.49 million tons in 2023 [9]. However, titanium slag can hardly be recycled because there are various impurities such as magnesium, aluminum, manganese, and titanium. Presently, only a small part of titanium slag is utilized to synthesize low-release fertilizer and sodium ferrate [10,11], while the rest is abandoned as an industrial solid waste and piled up like a mountain on industry premises. The accumulation of titanium slag not only takes up a massive land resource, but causes an enormous waste of resources. Meanwhile, some external factors such as rainfall leaching and wind erosion will lead to water and air pollution, which poses a potential risk to human and animal health. Hence, it is extremely important to propose a novel and effective method to recycle titanium slag as a secondary resource for sustainability in terms of replacing primary raw materials.

Hematite nanoparticles (HNPs) have attracted substantial attention because of their unique properties such as nontoxicity, chemical stability, biocompatibility, large surface area, and environmental benignity [12,13,14]. HNPs have been used extensively as batteries, fine ceramics, pigments, adsorbents, magnetic materials, and photocatalytic materials [15,16,17]. Currently, many technologies including hydrothermal, microemulsion, sol–gel, thermal decomposition, electrochemical deposition, and spray precipitation have been developed to synthesize HNPs [18,19,20]. Compared with the mentioned technologies, the pyrite (FeS2) reduction technology in this study is better suited for the formation of HNPs due to its easy operation, mild condition, simple technology, and no secondary pollution. Presently, HNPs have been fabricated using chemical reagents as feedstocks, which has restricted their industrial applications due to their high production costs. To reduce the production cost and enhance the utilization of titanium slag, titanium slag was used to synthesize HNPs as the iron source, which not merely turned the waste into valuable material, but also solved a large environmental problem. Moreover, sulfur dioxide released during the synthesis process was used to manufacture sulfuric acid, which attained the recycling of sulfur.

The primary objective of this study was to synthesize HNPs using titanium slag and pyrite as feedstocks by the reduction of titanium slag with pyrite under nitrogen protection. The effects of the synthesis conditions such as reaction temperature, reaction time, the mass ratio on the crystalline phase as well as crystalline size were systematically investigated by X-ray diffraction patterns. The reaction process was analyzed through thermogravimetry-differential scanning calorimetry. The physico-chemical properties of HNPs were investigated by multiple analytical techniques such as X-ray diffraction patterns, energy dispersive X-ray patterns, X-ray photoelectron spectroscopy, Fourier transform infrared spectra, scanning electron microscopy, ultraviolet–visible spectrophotometry, vibrating sample magnetometry, and the Brunauer–Emmett–Teller method.

## 2. Materials and Methods

### 2.1. Materials

Titanium slag composed of ferrous sulfate (FeSO_4_·7H_2_O, 92.5%) is an industrial by-product from titanium pigment production, which was supplied by the Titanium Industry of Pangang Group Corporation in Sichuan Province, China. Pyrite (FeS_2_, 72.6%) is an industrial by-product from mineral processing plants, and obtained from the Hanyuan Chemical Plant, Ya’an, Sichuan Province, China. Titanium slag (TS) and pyrite were used directly without any further treatment. Anhydrous ethanol was an analytical reagent that was purchased from the Kemiou Chemical Reagent Co. Ltd., Tianjin, China.

### 2.2. Thermodynamic Modelling

In this study, equilibrium compositions of the reactive system composed of FeSO_4_ and FeS_2_ at different temperatures were analyzed using the equilibrium composition modeling of HSC software prior to synthesizing the HNPs. The Gibbs free energy change in the corresponding reactions were estimated by reaction equations modeling in the HSC 6.0 software.

### 2.3. Fabrication of HNPs

Hematite nanoparticles (HNPs) were formed by the FeS_2_ reduction method using the as-received TS as the iron source and FeS_2_ as the reducing agent under a nitrogen atmosphere.

First, TS and pyrite without any further purification were separately vacuum dried at the temperature of 105 °C for 180 min, and then ground to pass through a 125 μm sieve in a ball mill. Subsequently, TS and pyrite, maintaining the mass ratio of TS to pyrite as 13:1–16:1, was well mixed in a corundum mortar. After that, the obtained mixtures were transferred into a porcelain boat and placed into a tube furnace (GSL-1500X, Hefei, China) with a programmed controlling-temperature. After being calcined at 550 °C for 30 min under nitrogen protection, the resultant samples were naturally cooled to ambient temperature and alternately rinsed with DI water and anhydrous ethanol according to the solid–liquid ratio of 0.5 g/mL to remove the soluble impurities. Finally, the as-resulted samples were dried at 80 °C for 240 min. In addition, sulfur dioxide (SO_2_) produced in the reaction was absorbed to be used to prepare sulfuric acid, the concentration of which was about 1–2 M due to the low emissions during the experimental process. The produced sulfuric acid was used to adjust then pH value of the solution. A flow sheet of the recycling hematite nanoparticles from titanium slag is depicted in Figure 1.

### 2.4. Characterization

Thermogravimetry-differential scanning calorimetry (TG-DSC, Netzsch, STA449F3, Bavaria, Germany) was used to investigate the reaction process under nitrogen protection. Identification of the crystalline phase for all the samples was conducted though X-ray diffraction (XRD, PANalytical B.V, Alemlo, The Netherlands) with a Cu-Kα radiation resource (λ = 1.5406 Å). Morphology and element composition were observed by scanning electron microscopy (SEM, Hitachi S4800, Tokyo, Japan) equipped with an energy dispersive X-ray pattern (EDX). Chemical state and elemental constitution for the sample were determined by X-ray photoelectron spectroscopy (XPS, Kratos, Manchester, UK) with monochromatic Al-Kα radiation. Surface performance was analyzed by Fourier transform infrared spectroscopy (FTIR, Nicolet 6700, Madison, WI, USA). The N_2_ adsorption–desorption isotherm at 77 K was applied to evaluate the specific surface area of the sample based on the Brunauer–Emmett–Teller method (BET, Asap2460, Mike, Atlanta, GA, USA) and pore size distribution of the sample according to the Barrett–Joyner–Halenda (BJH) method. The magnetic nature of the sample was measured with a magnetometer (VSM, Lakeshore7404, Westerville, OH, USA). The absorbance of the sample was scrutinized with an ultraviolet–visible spectrophotometer (UV–Vis, UV-3600, Shimadzu, Berlin, Japan).

## 3. Results and Discussion

### 3.1. Thermodynamic Analysis

Hematite nanoparticles were fabricated through the pyrite reduction approach using a reaction system consisting of FeSO_4_ and FeS_2_ as feedstocks under a nitrogen atmosphere. The equilibrium composition of FeSO_4_-FeS_2_ was analyzed by HSC software and the results are depicted in Figure 2A. From Figure 2A, the reactions in the FeSO_4_-FeS_2_ system began at 600 K and finished at 1000 K, and the main products formed in the reaction system were found to be Fe_2_O_3_, Fe_3_O_4_, and SO_2_. Therefore, it is inferred that there may be the following reactions of the FeSO_4_-FeS_2_ system at the temperature range of 600 K–1000 K.
(1)$$11FeSO_{4}+FeS_{2}\to 6Fe_{2}O_{3}+13SO_{2}$$
(2)$$8FeSO_{4}+FeS_{2}\to 3Fe_{3}O_{4}+10SO_{2}$$
(3)$$FeSO_{4}+Fe_{3}O_{4}\to 2Fe_{2}O_{3}+SO_{2}$$
(4)$$FeS_{2}+16Fe_{2}O_{3}\to 11Fe_{3}O_{4}+2SO_{2}$$

From Figure 2B, the Gibbs free energy change (∆*°G*) for reactions R1–R4 was less than zero at a temperature above 675 K and became more negative as the temperature rose, which indicates that the reaction between FeSO_4_ and FeS_2_ occurs at a temperature higher than 675 K, and increasing the temperature is beneficial to reactions R1–R4. However, the ∆*°G* of reaction R1 was more negative than those of reactions R2–R4 at the same temperature, showing that reaction R1 occurs more easily to form hematite at the same temperature. Hence, a single-product of hematite should be controlled to sustain the dominant reaction R1 and avoid side-reactions R2 and R4. From Figure 2A, hematite and magnetite were generated at 600–700 K, and the maximum amount of Fe_3_O_4_ reached and the reactant of FeS_2_ had been completely reacted at about 700 K, which indicated that reactions R1–R2 concurrently occur. Subsequently, equilibrium amounts of Fe_3_O_4_ and FeSO_4_ decreased gradually while those of Fe_2_O_3_ and SO_2_ increased until the equilibrium trended as the temperature rose to a temperature higher than 700 K, showing that FeSO_4_ reacts with the produced Fe_3_O_4_ to produce Fe_2_O_3_ and SO_2_, as shown in reaction R3. Hence, increasing with the appropriate content of FeSO_4_ is beneficial to enhancing the occurrence of reaction R3 and reduce the amount of Fe_3_O_4_. Based on the above-mentioned results, the reaction temperature and molar ratio of ferrous sulfate to pyrite are the two most significant factors in the preparation of hematite nanoparticles.

### 3.2. Reaction Process Analysis

The reaction temperature plays an important role in the synthesis of hematite nanoparticles. Therefore, the raw material before the reaction was investigated by simultaneous TG/DSC analysis, which operated from 40 °C to 800 °C at a heating rate of 10 °C/min under a nitrogen atmosphere. The TG/DSC curve of the reaction material is depicted in Figure 3. As indicated in Figure 3, three states of weight loss in the DTG/TG curve could be found as the temperature increased, and a total weight loss of approximately 52.87% was detected when the reaction material was heated up to 800 °C. The initial state below 250 °C with the mass loss of 4.40% can be assigned to released water adsorbed on the surface of the reaction material while the second state, which occurred within the temperature range from 250 °C up to 480 °C, had a mass loss of 9.98%, which corresponded to removing the crystal water in FeSO_4_·H_2_O [21]. The third state between 480 °C and 800 °C was caused by the reduction of TS with pyrite, which had a mass loss of 38.47%. When the temperature continued to rise to 800 °C, no obvious loss was detected. As can be seen from the above analysis, hematite nanoparticles were synthesized at a temperature above 480 °C due to the reduction of TS with pyrite. Similarly, the DSC curve of the mixture presented three corresponding endothermic peaks at approximately 196.6 °C, 337.0 °C, and 559.6 °C. The first endothermic peak of DSC curve at 196.7 °C can be attributed to releasing adsorbed water from the starting material. The second endothermic peak at 337.0 °C can be ascribed to the removal of the crystal water of FeSO_4_·H_2_O. The strong endothermic peak at 559.6 °C can be due to the reduction of TS (FeSO_4_) with pyrite, according to Equation (1). As a result, it can be inferred that hematite nanoparticles were synthesized by the mentioned reactions R1–R3.

### 3.3. Effect of Synthesis Conditions

The XRD patterns for all the materials fabricated at different temperatures are depicted in Figure 4A. From Figure 4A, the prominent peaks of the as-synthesized materials at a temperature above 550 °C were in conformity with those of the rhombohedral hematite (JCPDS 33-0664) with the R3c space group, and no additional peaks were discovered, which indicates that the as-synthesized nanometer materials were the pure phase of hematite. At the temperature of 500 °C, the characteristic diffraction peaks of magnetite (JCPDS 19-0629) at 2θ = 47.5° and 57.6° were discovered, except for the characteristic peaks of hematite (2θ = 24.0°, 33.1°, 35.6°, 40.8°, 49.4°, 54.0°, 62.4° and 64.1°), revealing that the fabricated materials belonged to the mixed-phase of hematite and magnetite. The results showed that the synthesized materials by the reduction of TS with pyrite were the pure phase of hematite at a temperature above 550 °C. In addition, the crystallite sizes were reckoned from the strongest peak of the XRD pattern by using Debye-Scherrer’s formula (DSF) at different temperatures, as indicated in Figure 4D. It can be clearly seen that the crystallite size increased as the reaction temperature increased. Hence, the optimal reaction temperature to fabricate HNPs by the pyrite reduction method is 550 °C.

The XRD patterns for all of the materials fabricated by the reduction of titanium with pyrite at different reaction times are displayed in Figure 4B. As indicated, the diffraction peaks of the synthesized materials detected at 2θ values of 24.0°, 33.1°, 35.6°, 40.8°, 49.4°, 54.0°, 62.4°, and 64.1° were respectively assigned to the reflection peaks of (012), (104), (110), (123), (024), (116), (214), and (300) lattice planes of hematite (JCPDS 33-0664), and no other impurity peaks were found at reaction times above 30 min, suggesting that the synthesized materials were the pure phase of hematite. While the new peaks appeared with 2θ values of 30.07° and 43.05° with a reaction time of 20 min can be ascribed to the (220) and (400) planes of magnetite with the space group of Fd3m (JCPDS 19-0629), revealing that the mixed-phase of hematite and magnetite formed at 20 min. Moreover, the crystallite sizes were estimated via using the Debye–Scherrer’s formula (DSF) base on the strongest peak of the XRD patterns, as presented in Figure 4B. It can clearly be seen that the crystallite size increased as the reaction time rose. Therefore, the best reaction time is 30 min to synthesize HNPs.

The XRD patterns of the synthesized materials at different mass ratios of TS to pyrite are presented in Figure 4C, in which all the detected diffraction peaks were well matched with the standard diffraction pattern of hematite (JCPDS 33-0664) without the presence of impurity peaks, indicating that the as-fabricated materials belonged to the pure phase of hematite at the mass ratio of 13:1–16:1. Beyond that, the smallest crystallite size of the HNPs estimated based on the (104) lattice plane was attained at a mass ratio of 14:1. Hence, the optimal proportion of TS to pyrite is 14:1.

### 3.4. Surface Property

FTIR spectroscopy was applied to investigate the surface properties of the synthesized materials by the pyrite reduction method, as shown in Figure 4E, in which the prominent peaks discovered at 544 and 467 cm^−1^ were separately ascribed to the Fe-O stretching vibration and bending vibration peaks in the HNPs [22,23], which is in accordance with the values reported in the literature whereas the two absorption peaks located at 3374 and 1638 cm^−1^ respectively corresponded to the O-H stretching vibration and bending vibration peaks of water molecules, which were adsorbed onto the surface of the HNPs [24,25]. Moreover, the characteristic band at approximately1103 cm^−1^ was attributed to the SO_4_^2−^ stretching vibration peak of FeSO_4_ in TS [26]. Therefore, the FTIR spectrum further verified that the as-formed nanoparticles were HNPs (α-Fe_2_O_3_).

### 3.5. Micrograph and Element Composites

The SEM image and EDX pattern of the HNPs formed through the pyrite reduction method are shown in Figure 5. The SEM image indicates that the prepared HNPs with a nearly spherical shape had a mean particulate size of ~45 nm, which was slightly larger than the value calculated from the XRD pattern (37.2 nm). This may have been due to the interparticle agglomeration and aggregation that resulted from electrostatic attraction, magneto–dipole interactions, and Van der Waals forces. Moreover, spherical nanoparticles with agglomeration and aggregation were obvious in the SEM image. The EDX pattern showed that the as-fabricated samples were composed of iron (58.39%), oxygen (36.68%), and carbon (4.64%) as well as sulfur elements. However, the content of sulfur was low, only 0.3%, and the appearance of the carbon peak was caused by a carbon copper grid applied in the process of analysis. In addition, the presence of iron and oxygen verified the main composition of the as-formed HNPs.

### 3.6. Surface Area and Pore Diameter

Nitrogen adsorption/desorption isotherms were separately employed to determine the specific surface area (SSA), pore size distribution (PSD), pore volume (PV) of the prepared HNPs via pyrite reduction approach at 550 °C. Figure 6A displays nitrogen adsorption/desorption isotherms on HNPs at 77 K. As mentioned, nitrogen adsorption–desorption isotherms present adsorption hysteresis loop without showing restrictive adsorption at high relative pressure. Thus, the nitrogen adsorption–desorption isotherms of the HNPs demonstrated type-IV with a H3 hysteresis loop base in the IUPAC classification [27], showing the characteristic of mesoporous materials with a slit-shaped pore, which can be caused by aggregates of particles. According to the nitrogen adsorption/desorption isotherm, the BET surface area, BJH pore volume, and pore diameter of the fabricated HNPs were obtained as 19.6 m^2^/g, 0.117 cm^3^/g, and 0.89 nm, respectively. Hence, the results show that the as-fabricated material had a mesoporous structure.

### 3.7. Magnetism and Absorbance of Material

The analysis of the magnetic properties for the formed HNPs is of great significance to practical applications. To clarify the magnetic performance, the room temperature magnetization curve was performed by using a vibrating sample magnetometer within an applied magnetic field range from −10 kOe to 10 kOe. The magnetization curve of HNPs is displayed in Figure 6B. From Figure 6B, the magnetization curve for HNPs revealed an S-shaped hysteresis (M-H) loop with low remnant magnetization (Mr = 1.3 emu/g) and weak coercivity (Hc = 0.32 kOe), indicating that the formed HNPs exhibited typical ferromagnetic behavior. Additionally, the magnetization value approached saturation at the applied magnetic field of 8500 kOe, and the saturation magnetization (Ms) value estimated from the magnetic hysteresis (M-H) loops was ca. 5.15 emu/g, which was substantially higher than the values documented for hematite [28]. The high Ms value could be a result of form surface spin disorder [29], which is more easily aligned in the direction of the magnetic field than core spins [30]. The above-mentioned results show that the as-formed HNPs possess weak ferromagnetic behavior at room temperature.

The optical performance of the prepared HNPs is crucial to their application in photocatalysis. Hence, the absorbance of the prepared HNPs via the pyrite reduction method was investigated using an ultraviolet–visible spectrophotometer, as shown in Figure 6C. It can be clearly seen that the absorption threshold of the fabricated HNPs was located at around 600 nm and that they had a strong absorption light capacity within the wavelength range of 200 nm–600 nm, indicating that HNPs have good absorbance in the ultraviolet (200 nm–400 nm) and visible light region (400 nm–600 nm). Compared with the values of HNPs reported in the literature [31], the resulting HNPs had stronger light absorption capacity in the ultraviolet and visible light region [32], which could be due to impurities (Ti, Mg, Al) in the HNPs from TS [33,34]. Hence, the resulting HNPs had good absorbance in the visible light region, which were used as the visible light responsive catalyst.

### 3.8. XPS Analysis and Chemical State

The oxidation state and chemical compositions of the fabricated HNPs were investigated by XPS, as presented in Figure 7. The six prominent peaks of the XPS wide-scan spectrum at 722.8, 709.5, 529.81, 401.81, 284.81, and 167.81 eV separately correspond to the Fe2p_1/2_, Fe2p_3/2_, O1s, N1s, C1s, and S2p core level spectra [35]. The XPS spectrum revealed that the as-formed material was mainly composed of iron and oxygen. Moreover, the Fe2p_3/2_ and Fe2p_1/2_ peaks located at 722.8 and 709.5 eV are one of the major characteristics of hematite (Figure 7B) [36]. In addition to this, a weak satellite peak observed at 717.0 eV is a typical feature of hematite, revealing the formation of HNPs [37,38]. The O1s core level of HNPs can be deconvoluted into three peaks, which corresponded to the binding energies of 529.6, 531.4, and 533.3 eV (Figure 7C). The obvious peak at 529.6 eV was assigned to lattice oxygen of HNPs, whereas the characteristic peak at the binding energy of 531.4 eV was attributed to the presence of O-H on the surface of the HNPs. The C-O peak observed at 533.3 eV may have been caused by the CO_2_ loaded on the HNPs from air [39]. The C1s region can be fitted into three peaks at the binding energies of 284.6, 285.8, and 288.4 eV (Figure 7D), which were ascribed to the lattice C (used as reference), C-O, and S=O functional groups [19,40]. Therefore, these results further confirm that the surface of the synthesized nanometer materials contain Fe-O and O-H functional groups.

### 3.9. Recovery of Metal Ions in Washing

The concentration of Zn, Mg, Mn, Fe(II) in washing was less than 0.8 g/L. The main methods to recover low concentrations of Zn, Mg, Mn, Fe(II) from aqueous solutions or industrial wastewater are membrane separation, ion exchange, and adsorption [41,42]. Azadian et al. [43] used the synthesized engineered biochar as an adsorbent for recycling Zn(II) from an aqueous solution, while Pourshadlou et al. [44] used a bentonite/gamma-alumina nanocomposite as an adsorbent for adsorbing Mg(II) from an aqueous solution. Soheil et al. [45] applied the synthesized nanofiltration membranes by the phase inversion method for recovering Zn(II) and Pb(II) from industrial wastewater. Ion exchange was employed to remove Fe(II), Al, Co, and Mn ions from the leach solution [46]. Among the above-mentioned methods, membrane technology has wide applications in recycling metal ions in solution because of its many advantages such as large-scale, continuous separation, and is environmental-friendly. Therefore, membrane technology will be used to recycle Zn, Mg, Mn, Fe(II) from aqueous solutions from washing in future study.

## 4. Conclusions

Hematite nanoparticles (HNPs) were successfully fabricated by the pyrite reduction method under a nitrogen atmosphere. The results verified that the optimal conditions for the formation of HNPs were 550 °C for 30 min with the mass ratio of 14:1 for TS and pyrite, respectively. The average particulate size of the HNPs with a sphere-like structure observed from the SEM image was approximately 45 nm. The specific surface area, pore volume, and pore size of the HNPs with a mesoporous structure were respectively about 19.6 m^2^/g, 0.117 cm^3^/g, and 0.89 nm. In addition, the XPS analysis further confirmed that the resulting materials were the pure phase of hematite, and the O/Fe in HNPs was close to stoichiometric hematite. The magnetization curve at room temperature demonstrates that the resulting HNPs possess weak ferromagnetic behavior. The as-resulted HNPs investigated using the UV–Vis spectrum also had good absorbance in both the ultraviolet region (200–400 nm) and visible light region (400–600 nm) and can be applied as a visible light responsive catalyst. Consequently, the results indicate that hematite nanoparticles fabricated by the pyrite reduction method have promising application prospects in adsorbents, magnetic materials, and photocatalysis.

## Figures and Tables

**Figure 1 nanomaterials-14-01330-f001:**
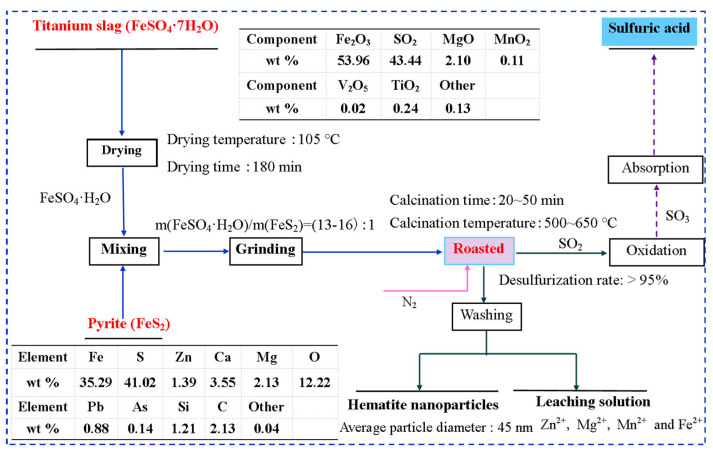
Flow sheet of recycling hematite nanoparticles from titanium slag.

**Figure 2 nanomaterials-14-01330-f002:**
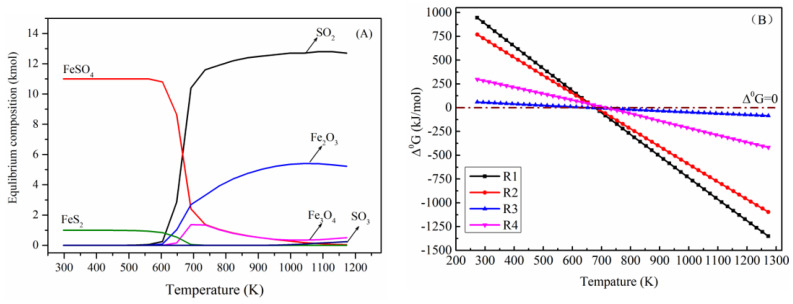
(**A**) Equilibrium composition of the FeS_2_-FeSO_4_ system and (**B**) Gibbs free energy change under different temperatures.

**Figure 3 nanomaterials-14-01330-f003:**
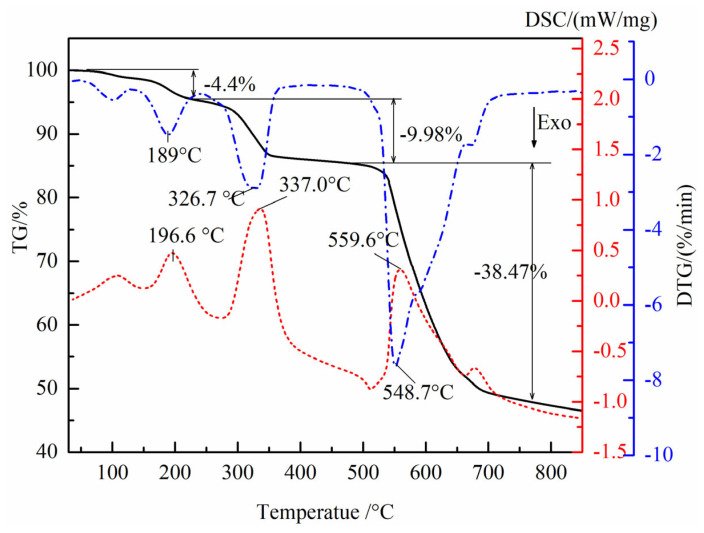
TG/DSC curves of the mixture of TS and pyrite (14:1).

**Figure 4 nanomaterials-14-01330-f004:**
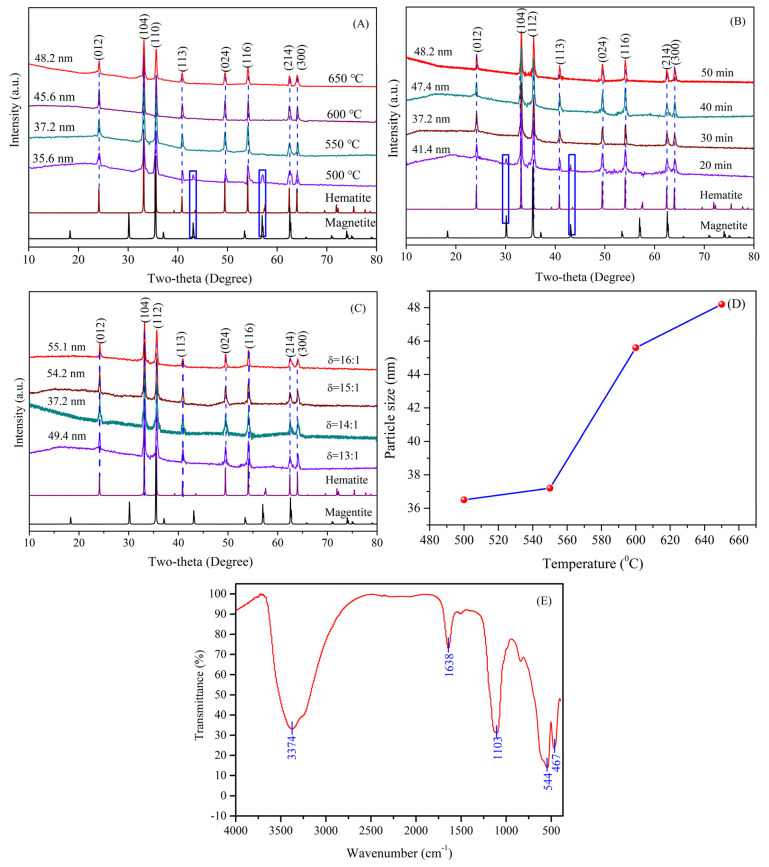
X-ray diffraction patterns: (**A**) different reaction temperatures, (**B**) different reaction times, (**C**) different ratios, (**D**) particle size distribution at different temperatures, and (**E**) FTIR spectrum for HNPs.

**Figure 5 nanomaterials-14-01330-f005:**
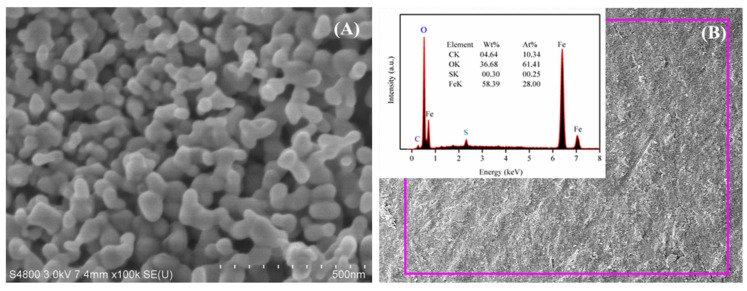
(**A**) SEM image and (**B**) EDX pattern of the HNPs.

**Figure 6 nanomaterials-14-01330-f006:**
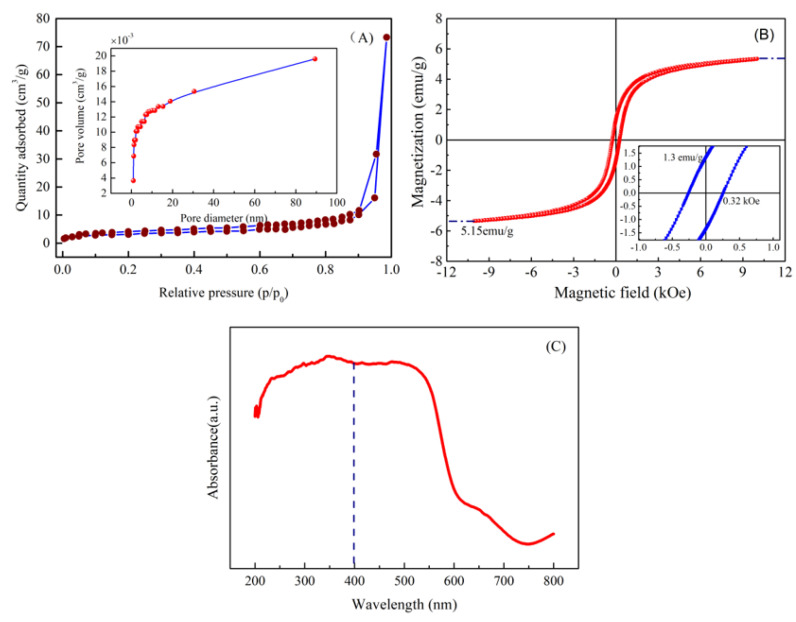
(**A**) N_2_ adsorption/desorption isotherm, (**B**) field dependent magnetization hysteresis loops, (**C**) UV–Vis absorbance spectrum of HNPs.

**Figure 7 nanomaterials-14-01330-f007:**
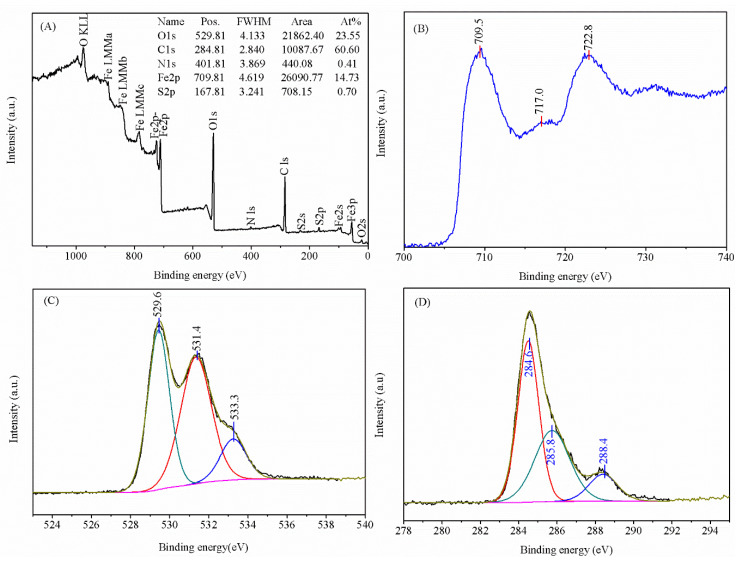
The XPS spectra of HNPs: (**A**) wide-scan, (**B**) Fe2p, (**C**) O1s, and (**D**) Cls.

## Data Availability

Data will be made available on request.

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
