# Peer review of "A Novel Fabrication of Hematite Nanoparticles via Recycling of Titanium Slag by Pyrite Reduction Technology"

_nanomaterials, 2024, doi:10.3390/nano14161330_

Round 1

Reviewer 1 Report

Comments and Suggestions for Authors

Manuscript ID: nanomaterials-3116287

Title: A novel fabrication of hematite nanoparticles via recycling of titanium slag by pyrite reduction technology

Authors: Genkuan Ren et al.

Keywords: “waste utilization” should be added.

Line 32, 41. Authors should add information about 2023 year in China.

Line 54, 55, 58. Avoid more than 3 references for a fact in one sentence. A maximum of 3 in a sentence is allowed for Nanomaterials. Describe this information in detail.

Line 96. What the volume of water and ethanol used for washing?

Authors should add the section about thermodynamic modelling to the section 2.

What are the changes in the particle size distribution (nm) of the samples after roasting in the T = 500-650 °C? Add the LD figure.

What is the chemical composition (wt.%; impurities) of the final HNPs powder? What is the S content?

Add the information about sulfuric acid concentration (M or %) after absorbtion.

What is the Zn, Mg, Mn, Fe(II) content (g/L) in the solution after washing? Add to discuss with references of the future recycle of this solution.

Authors should compare the HNPs powder with the China State standard (中国国家标准) at the by physical properties and chemical composition.

Author Response

Thank you very much for taking the time to review this manuscrpit. your advice .Your suggestions are helpful to improve the quality of my manuscript to a great extent. thank you 

Reviewer 2 Report

Comments and Suggestions for Authors

An article on methods of producing hematite nanoparticles by releasing pyrite waste in a nitrogen environment. The text has a corrected product name, which comes from scientific works, and therefore contains a description of the problems, the procedures and universal methods used, the use of broad analytical methods for the newly produced material, critically described in relation to the subject literature and summaries of research results. The list of literature contains the latest research on the subject of the problem and is international, including foreign works. I think that the research problem is interesting and presented in detail. I recommend publication of the article with one comment: references to literary items in the text should be in the form of normal numbers and not as a superscript.

Author Response

(The authors gave the same response as above.)

Reviewer 3 Report

Comments and Suggestions for Authors

Dear Authors,

Congratulations for your work!

Generally - interesting and useful paper, aimed at using two industrial wastes to produce valuable material. The use of pyrite and pyrite waste as a reducing agent is known in the scientific community, but its use to reduce iron compounds from the production of titan-based products is not well studied. Experiments are well thought out, based on thermodynamic calculations and TG/DSC curves analysis further decisions on process conditions are made. The influence of various factors that affect the quality of the obtained hematite nanoparticles was investigated. The synthesized nanoparticles are well characterized using modern methods - XRD, SEM-EDX, FTIR, XPS, N2 adsorption/desorption (BET and BJH methods) UV-vis, VSM.

To me it is a bit "strong" to claim that the synthesized material would be a good adsorbent based solely on data on nitrogen adsorption/desorption isotherms without any adsorption experiments.

The work would benefit from even a small comment on the possible impurities of other metals in the formed nanoparticles, as they could affect the potential use of the resulting nanoparticles.

There is a need for small technical corrections - in my opinion, the main reason for the incorrect use of terms is the use of the English language. In order to facilitate the authors, I am attaching a file of the article with comments and suggestion inside it.

Sincerely,

Comments on the Quality of English Language

Moderate corrections are needed - plural / singular are not always correctly used and most - important some scientific terms are not correct

Author Response

Dear Reviewer

Thank you very much for taking the time to review this manuscrpit. your advice .Your suggestions are helpful to improve the quality of my manuscript to a great extent. Thank you for review the manuscript. We  have revised the manuscrpit according to your advices

Round 2

Reviewer 1 Report

Comments and Suggestions for Authors

The authors have significantly improved the article and added a new section 3.9 with links. I believe that this version of the article is of a much higher quality and can be accepted in this form.